# Sight Beyond Text: Multi-Modal Training Enhances LLMs in Truthfulness and Ethics

**Haoqin Tu**[1]*, **Bingchen Zhao**[2]*, **Chen Wei**[3], **Cihang Xie**[1]

[1]University of California, Santa Cruz      [2]University of Edinburgh      [3]Rice University

**Reviewed on OpenReview:** <https://openreview.net/forum?id=2Zl0zc7fO8>

## Abstract

Multi-modal large language models (MLLMs) are trained based on large language models (LLM), with an enhanced capability to comprehend multi-modal inputs and generate textual responses. While they excel in multi-modal tasks, the conventional view within the machine learning community has often undervalued/overlooked their capabilities in pure natural language processing. This paper aims to get out of the box and showcase an intriguing characteristic of multi-modal trained LLMs — our preliminary results suggest that visual instruction tuning, a prevailing strategy to integrate vision knowledge into the LLMs, unexpectedly and interestingly helps models attain both improved truthfulness and ethical alignment in the pure NLP context. For example, a visual-instruction-tuned LLaMA2 7B model surpasses the performance of the LLaMA2-chat 7B model, fine-tuned with over one million human annotations, on `TruthfulQA` and `Ethics` benchmarks. Similarly, the latest LLaMA3 series also shows consistent performance gains by 0.6% on average following visual-instruction tuning. Another example is that two versions of proprietary model GPT-4V-turbo, which incorporates visual information, surpasses its LLM-only counterpart GPT-4-turbo by around 1.6% on both aspects. Further analysis reveals that the improved alignment can be attributed to the superior instruction quality inherent to visual-text data. By presenting those findings, we advocate for a broader exploration into visual-text synergies, positing that such multi-modal interactions could be pivotal in advancing alignment research. In releasing our code at <https://github.com/UCSC-VLAA/Sight-Beyond-Text>, we aspire to foster further exploration into the intrinsic value of visual-text synergies and, in a broader scope, multi-modal interactions in alignment research.

## 1 Introduction

Enhancing truthfulness and reducing hallucinations of Large Language Models (LLMs) is one the paramount challenges in the domain of artificial intelligence. This paper introduces a new perspective on this research topic, advocating for the integration of multi-modal data into LLM training as a strategy to significantly improve their truthfulness and alignment with human values.

Our stance is informed by empirical evidence demonstrating the beneficial impact of diverse data sources on LLM capabilities. For example, the inclusion of code data has been shown to improve the reasoning ability of LLMs (Ma et al., 2024). Building upon this premise, this paper aims to explore the potential benefits of an even more diverse data source – multi-modal data, particularly images, in enhancing the capabilities of LLMs.

Most modern MLLMs leverage LLMs as their core, setting the aim to bridge the gap between language and visual tokens (Liu et al., 2023b; Li et al., 2023a; Ye et al., 2023). While language tokens often capture much of the real-world context, visual information is essential to share richer real-world details that connect to the factual knowledge in human experience (Harnad, 1990; Bisk et al., 2020; Tu et al., 2023b), particularly in

---

*Equal contribution.

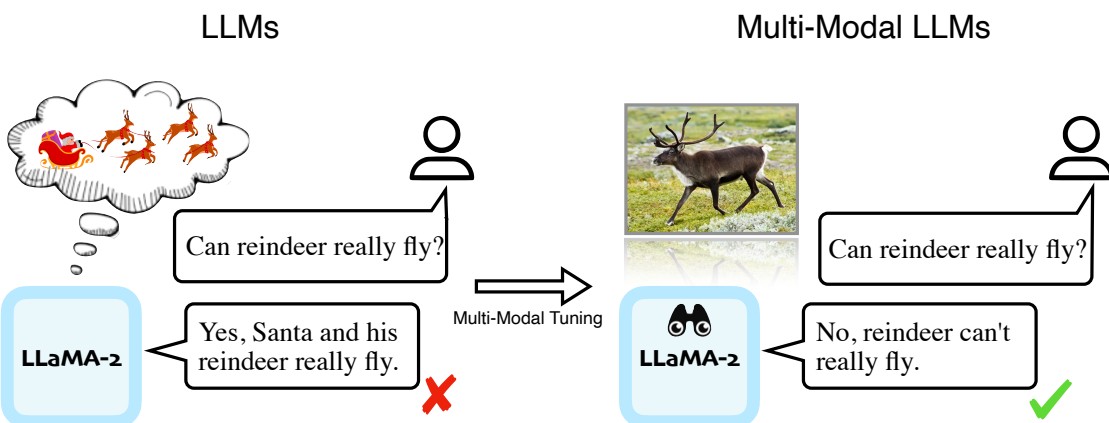

Figure 1: Visual instruction tuning substantially improves the truthfulness and ethics of LLMs. We observe tuning LLMs with only 80k multi-modal data can yield stronger results on truthfulness and ethics than those with over one million human-annotated RLHF data. Note that these LLMs employ images only during the visual instruction tuning and are tested without images for NLP tasks.

areas related to truthfulness and ethics. Consequently, guiding LLMs to integrate and process visual tokens enhances the model's performance in dimensions such as ethics and truthfulness. Our claim is firmly grounded in our experimental evidence. In our preliminary explorations, we tune LLaMA series models (Touvron et al., 2023a;b) with the visual instruction data from LLaVA (Liu et al., 2023b;a). The results of these experiments are intriguing: for a vanilla LLaMA2 7B model, visual instruction tuning can register impressive scores of 46.0% on `TruthfulQA-mc` (+7.1%) (Lin et al., 2022) and 65.4% on `Ethics` (+19.6%) (Hendrycks et al., 2020), depending on the specific tuning approach. It is particularly noteworthy that, even without engineering efforts that explicitly elicit ethical or truthful behaviors, the performance of the visual instruction-tuned model already outperforms that of the LLaMA2-chat 7B variant, which is heavily tuned with over a million human annotations (Touvron et al., 2023b).

We hypothesize that, in real-world scenarios, visual signals can enhance language models (LMs) in three key ways: by providing enhanced contextual grounding through explicit clues (*e.g.*, settings, object relationships, and implied actions), integrating implicit knowledge that, while not explicitly stated in the text, is crucial for ethical decision-making, and mitigating bias by incorporating diverse, real-world depictions that reflect a broader range of human experiences and cultural practices.

In proposing this novel perspective, we aim to spur a possible paradigm upgrade or even a complete shift to the ongoing dialogue within the machine learning community. We contend that broadening the data diversity for LLMs, beyond traditional text-based inputs, is a pivotal step towards developing models that more accurately reflect, interpret, and respond to the complexities of real-world information (Ma et al., 2023). This paper seeks to engage the community in a discussion about this evolving approach, underlining its potential impact on the ethical and responsible aspects of AI development.

In summary, our insights accentuate the promise of visual instruction tuning in fostering the ethical and truthful alignment of LLMs. It is our hope that this paper will serve as a catalyst for a new wave of research, one that embraces the rich possibilities offered by multi-modal data and paves the way for more aligned and responsible AI systems.

## 2 Tuning LLMs with Multi-Modal Data

This section introduces our strategies to tune LLMs using multi-modal datasets. A standard MLLM typically contains three key components: 1) a vision encoder tasked with encoding visual inputs, 2) a vision-language connector that translates visual tokens into the linguistic space, and 3) an LLM for decoding the transcribed

visual information. We strictly adhere to the setups in LLaVA (Liu et al., 2023b) for fine-tuning LLMs on visual instruction tuning data.

**Model Architecture.** We incorporate the pre-trained visual branch of CLIP ViT-L/14 (Radford et al., 2021) as our vision encoder. Additionally, a trainable linear layer is employed to project visual tokens into the language embedding space. Regarding the choice of LLM, we take the widely recognized open-sourced LLaMA models (Touvron et al., 2023a;b; Geng & Liu, 2023) for this study. Specifically, our investigation focuses on the following six models, containing three latest LLMs and their corresponding instruction-tuned variants:

- Pre-trained LLM: OpenLLaMA-3B (Geng & Liu, 2023), LLaMA-7B (Touvron et al., 2023a), LLaMA2-7B (Touvron et al., 2023b).

- Instruction-tuned LLM: OpenAlpaca-3B (Su et al., 2023), LLaMA2-chat-7B (Touvron et al., 2023b), the Vicuna family (Vicuna-7B, Vicuna-v1.5-7B, Vicuna-v1.5-13B) (Zheng et al., 2023), the LLaMA-3-Instruct family (LLaMA3-8B, LLaMA-3.1-8B, LLaMA-3.2-11B) (Dubey et al., 2024), the OpenAI's GPT-4 family (GPT-4-turbo, GPT-4V-turbo) (Achiam et al., 2023a;b).

As listed above, our study is centered on two model scales: 3B and 7B. While the 3B LLaMA model is sourced from the OpenLM project (Geng & Liu, 2023), the 7B LLaMA models are directly released by Meta (Touvron et al., 2023a;b); additionally, our investigation extends to the instruction-tuned variants of these base LLMs. Concretely, OpenAlpaca-3B is fine-tuned on the Alpaca data (Taori et al., 2023) using OpenLLaMA-3B as its backbone; Vicuna-7B is the v1.1 model from FastChat (Zheng et al., 2023), which is crafted upon LLaMA-7B and employs 125K conversational data from ShareGPT (ShareGPT) during tuning; LLaMA2-chat-7B is well-engineered for human alignment, undergoing its training on publicly available instruction datasets and one million human-annotated examples using RLHF techniques. The LLaMA3 series is the most recently released open-weight model family with powerful NLP abilities. For LLaMA3.2-11B, we only take the LLM part in the model for experiments. Note that we test 7B/8B LLM variants by default, and indicate 3B models by the suffix "-3B".

**Training Procedure.** The MLLM training unfolds in two stages. First, we exclusively tune the weight of the vision-language connector, with both the visual encoder and the LLM remaining frozen. In the second phase, we fine-tune the weights of both the connector and the LLM. Data-wise, we adhere to the protocols set by LLaVA (Liu et al., 2023b): the connector is initially trained using 595k image-text pairings filtered from CC3M (Changpinyo et al., 2021); the subsequent stage that requires LLM training utilizes 158k instructions-following data from LLaVA with 80k unique images, which contains image-grounded conversation, image descriptions, and image-based complex reasoning tasks. To investigate the factors driving the improvements in visual instruction tuning, we also explore tuning the model using only text-based instruction data. We utilize three types of text-only data (sampled to equal sizes): visual instruction tuning data without images, Alpaca data (Taori et al., 2023), and Orca data (Lian et al., 2023). The Alpaca dataset is derived from prompting OpenAI's GPT model with a variety of real-world questions and scenarios, while the Orca dataset comprises FLAN-augmented examples (Longpre et al., 2023), which have been shown to empower open-source 13B LLMs to excel across multiple benchmarks. (Lian et al., 2023). As for the training strategy, we probe the effects of both full fine-tuning and LoRA fine-tuning (Hu et al., 2021).

**Evaluation Protocols.** We conduct our evaluation using a publicly available and widely used pipeline (Gao et al., 2021). Specifically, for the `Ethics` benchmark, we use accuracy as the evaluation metric. For `TruthfulQA`, we follow the official repository and use Rouge and/or BLEU accuracy for generation tasks, along with single-true (mc1) and multi-true (mc2) metrics for question-answering. Other NLP tasks are evaluated with accuracy or F1 score, following the original work. For multi-modal tasks, we use accuracy for question-answering benchmarks and CIDEr (Vedantam et al., 2015) for generation tasks. Further details on multi-modal benchmark metrics will be presented in Section 3.4.

| Models | Ethics Acc. | TruthfulQA-gen Rouge Acc. | TruthfulQA-mc1 Acc. | TruthfulQA-mc2 Acc. |
|---|---|---|---|---|
| Delphi | 60.1% | - | - | - |
| TA (LLaMA2-chat) | - | - | - | 45.2% |
| TA (Vicuna-v1.5-7B) | - | - | - | 50.0% |
| LLaMA-7B | 50.4% | 27.5% | 22.0% | 34.1% |
| MM-ft | 59.1% (+8.7%) | 29.4% (+1.8%) | 23.6% (+1.6%) | 35.8% (+1.7%) |
| LLaMA-3B | 45.6% | 25.3% | 21.3% | 34.6% |
| MM-ft | 58.1% (+12.5%) | 26.4% (+1.1%) | 21.4% (+0.1%) | 32.9% (-1.7%) |
| MM-lora | 45.7% (+0.1%) | 25.2% (-0.1%) | 23.0% (+1.7%) | 35.6% (+1.0%) |
| Alpaca-3B | 44.0% | 28.6% | 22.4% | 34.2% |
| MM-ft | 46.8% (+2.8%) | 28.2% (-0.4%) | 23.1% (+0.7%) | 34.2% (+0.0%) |
| MM-lora | 44.0% (+0.0%) | 28.6% (+0.0%) | 24.6% (+2.2%) | 38.0% (+3.8%) |
| LLaMA2 | 45.8% | 32.3% | 25.2% | 38.9% |
| MM-ft | 65.4% (+19.6%) | 31.5% (-0.9%) | 27.8% (+2.6%) | 40.2% (+1.3%) |
| MM-lora | 46.1% (+0.3%) | 37.9% (+5.6%) | 32.1% (+6.9%) | 46.0% (+7.1%) |
| LLaMA2-chat | 58.5% | 43.3% | 29.5% | 44.6% |
| MM-ft | 65.2% (+6.7%) | 35.5% (-7.8%) | 27.7% (-1.8%) | 41.0% (-3.6%) |
| MM-lora | 58.6% (+0.1%) | 44.6% (+1.2%) | 29.4% (-0.1%) | 44.6% (+0.0%) |
| Vicuna-v1.5-7B | 62.1% | 40.6% | 28.9% | 45.4% |
| MM-ft | 69.1% (+7.0%) | 40.8% (+0.2%) | 30.8% (+1.9%) | 45.9% (+0.5%) |
| MM-lora | 64.5% (+2.4%) | 44.6% (+4.0%) | 31.8% (+2.9%) | 47.6% (+2.2%) |
| Vicuna-v1.5-13B | 69.1% | 39.0% | 30.1% | 45.9% |
| MM-ft | 73.9% (+4.8%) | 38.8% (-0.2%) | 29.4% (-0.7%) | 42.5% (-3.4%) |
| MM-lora | 69.8% (+0.6%) | 42.2% (+3.2%) | 29.4% (-0.7%) | 46.3% (+0.4%) |
| LLaMA3-8B | 67.6% | 47.7% | 36.1% | 51.5% |
| MM-ft | 68.1% (+0.5%) | 47.3% (-0.4%) | 36.7% (+0.6%) | 51.5% (-0.5%) |
| MM-lora | 67.4% (-0.2%) | 48.3% (+0.6%) | 36.8% (+0.7%) | 51.9% (+0.4%) |
| LLaMA3.1 | 66.9% | 60.6% | 36.7% | 54.0% |
| LLaMA3.2* | 68.5% (+1.6%) | 63.0% (+2.4%) | 37.7% (+1.0%) | 54.9% (+0.9%) |

Table 1: Comparison on the original LLMs and the multi-modal fine-tuned ones on Ethics (Hendrycks et al., 2020) and TruthfulQA (Lin et al., 2022). '-ft' represents full parameter fine-tuning and '-lora' indicates LoRA tuning. We report Rouge-L accuracy for TruthfulQA-gen and accuracy for the rest. Note that LLaMA3.2* denotes a model with visual capabilities built on LLaMA3.1. Note that we take the Delphi (Jiang et al., 2021) and the Trustworthy-Alignment (TA) (Zhang et al., 2024) to provide baseline numbers for these two tasks.

# 3 Evaluations Results and Analysis

## 3.1 Truthfulness and Ethics of MLLMs

We report the evaluation results on the TruthfulQA and Ethics benchmarks, designed for measuring LLMs' truthfulness and ethical alignment. During this evaluation, we utilize the weights exclusively from the visual-instruction-tuned LLMs, intentionally omitting the visual encoders and vision-language connectors introduced during the fine-tuning process. The results are presented in table 1.

**Visual Instruction Tuning Improves Truthfulness and Ethics.** Our observations suggest that, rather unexpectedly, visual instruction tuning tends to enhance the truthfulness of LLMs. A compelling observation emerges when comparing between LLaMA2 and LLaMA3 variants: visual-instruction-tuned models, especially LLaMA2 with MM-lora, surpass the LLaMA2-chat model in performance metrics on both TruthfulQA-mc1 (32.1% *vs.* 29.5%) and TruthfulQA-mc2 (46.0% *vs.* 44.6%). Furthermore, as one of the leading open-weight

| Data | Ethics
Acc. | TruthfulQA
BLEU Acc. | TruthfulQA
Rouge Acc. |
|---|---|---|---|
| GPT-4-turbo (0515) | 87.0% | 55.0% | 55.3% |
| GPT-4V-turbo (0515) | 87.4% (+0.4%) | 55.5% (+0.5%) | 54.9% (-0.4%) |
| GPT-4-turbo (1201) | 87.3% | 54.0% | 54.8% |
| GPT-4V-turbo (1201) | 88.0% (+1.3%) | 54.6% (+0.6%) | 54.4% (-0.4%) |

Table 2: Comparison on GPT-4-turbo and its multi-modal variant GPT-4V-turbo on `Ethics` and `TruthfulQA` generation. We report results of GPT-4-turbo models with different timestamps (*e.g.*, 0515, 1201). For `TruthfulQA`, we use BLEU and Rouge-L accuracy on its generation task.

LLMs, the LLaMA3 series shows noticeable performance gains when visual instruction data is integrated, with average improvements of 0.6% and 0.9% on the `Ethics` and `TruthfulQA-gen`, respectively.

From table 1, we also observe visual instruction tuning leads to substantial improvements on the `Ethics` task. Echoing the trend in the `TruthfulQA` evaluations, visual-instruction-tuned models, specifically the MM-ft versions of both LLaMA2, consistently outpace their instruction-tuned counterparts, such as LLaMA2-chat and Alpaca-3B. For example, the performance enhancements observed for LLaMA2 on the `Ethics` task amounted to increments of 19.6%, outperforming LLaMA2-chat and Alpaca-3B by margins of 6.9% and 11.3%. Another straightforward observation is that, models with larger parameter scale generally perform better in these two aspects (*e.g.*, 13B *vs.* 7B *vs.* 3B LLMs), as stronger base LLMs are more capable during the multi-modal tuning process.

For more recent LLMs like the Vicuna-v1.5 and LLaMA3 family, we also have the observation that the visual-instruction-tuned MLLMs perform better than its language-only counterparts by 2.2% and 0.8% across all Vicuna-v1.5 and LLaMA3 models on Ethics and TruthfulQA, respectively. While finetuning the LLM part in MLLM with full parameter activation leads to better multi-modal performance, which has also been validated by other works, it generally underperforms LoRA-tuned MLLMs in TruthfulQA task (*i.e.*, 2.2%, 2.4%, and 2.6% improvement of LoRA tuning compared with finetuning for Vicuna-1.5-7B, Vicuna-1.5-13B, LLaMA-3.1, respectively). But LoRA-tuned MLLM lags behind finetuned ones on Ethics by an average of 7.2% across 7 model variants. This observation suggests that the ethics aspect aligns better with the multi-modal objective in visual instruction tuning than the truthfulness aspect as discussed in Section 3.1.

For recent LLMs like Vicuna-v1.5 and the LLaMA3 family, we observe that visual-instruction-tuned MLLMs outperform their language-only counterparts, with gains of 2.2% and 0.8% on the Ethics and TruthfulQA benchmarks, respectively, across all Vicuna-v1.5 and LLaMA3 models. While finetuning the full LLM component in MLLMs enhances multi-modal performance (Liu et al., 2023b; Li et al., 2023a)—it generally underperforms compared to LoRA-tuned MLLMs on the TruthfulQA task. Specifically, LoRA tuning yields improvements of 2.2%, 2.4%, and 2.6% over finetuning for Vicuna-1.5-7B, Vicuna-1.5-13B, and LLaMA-3.1, respectively. However, LoRA-tuned MLLMs fall behind finetuned ones by an average of 7.2% on the Ethics benchmark across seven model variants. This suggests that alignment with the multi-modal objective in visual instruction tuning may be stronger for ethics-related dimensions than for truthfulness, we will dive into this aspect and discuss more later.

It should be noted that the employed visual instruction tuning data (that requires LLM parameter update) is the 158k dataset derived from LLaVA (Liu et al., 2023b), **which does not contain special designs for aligning models to human preferences.** Remarkably, despite this, visual instruction tuning is able to yield empirical advantages that surpass those from RLHF, which heavily utilizes a substantial corpus of human-annotated data dedicated to LLM alignment. This observation strongly attests to the potential that visual instruction tuning holds in addressing AI alignment challenges.

However, it is not a silver bullet — our experiments also show that visual instruction tuning is limited at enhancing the alignment of models previously fine-tuned via instruction tuning (*e.g.*, models like Vicuna, LLaMA2-chat), indicating variability in its efficacy.

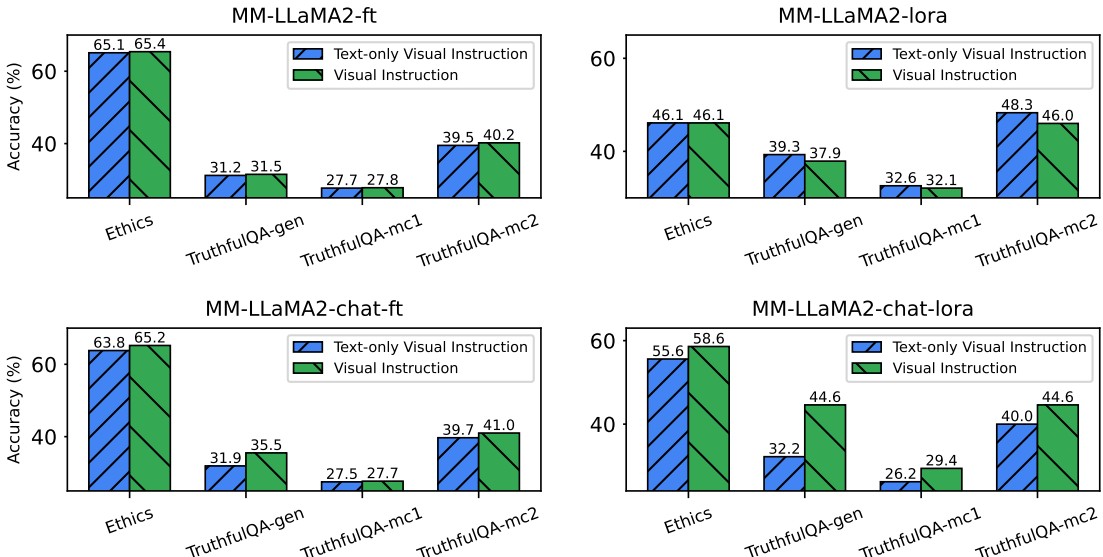

Figure 2: Performance of visual-instruction-tuned LLaMA2 models and text instruction tuned ones on `Ethics` and `TruthfulQA` benchmarks. The text-only visual instruction data is taken directly from LLaVA, but without the paired images.

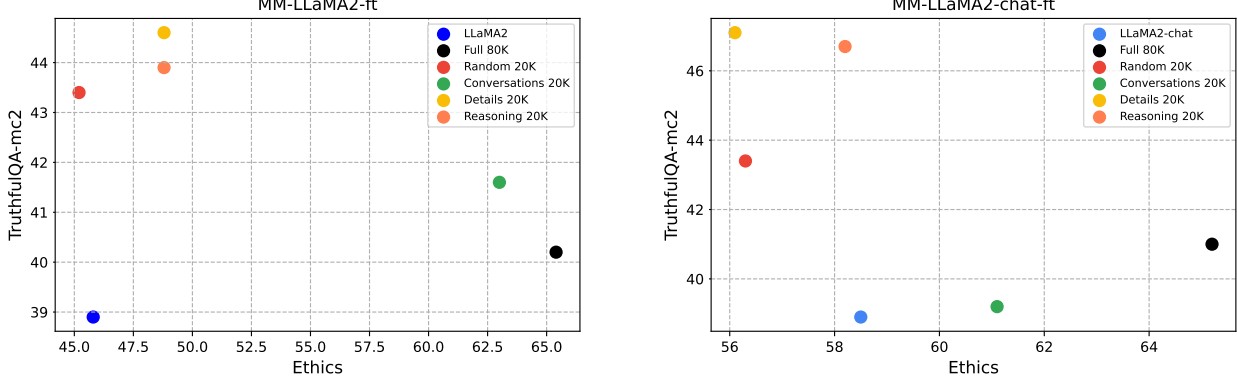

Figure 3: Results of different data components on `Ethics` and `TruthfulQA` of visual-instruction-tuned LLMs. We utilize 20K of different forms of data (Conversation, Details, Reasoning), and additionally sample 20K data out of the original 80K training instances (Random 20K) for comparison.

We also report model performance of two versions of proprietary GPT-4-turbo series on these two NLP tasks in table 2. The GPT-4V model is regarded as an upgrade of GPT-4-turbo, with the visual understanding ability. The GPT-4V demonstrates improved performance on `Ethics` by an average of 0.9%, as well as on the `TruthfulQA` generation task under BLEU accuracy, further supporting our claim in bringing visual knowledge to enhance LLMs' ethical and truthful awareness.

**Effects of Modalities in Visual Instruction-Tuning Data on LLM Alignment.** Next, we seek to understand how different modalities in the visual instruction data contribute to the alignment of LLMs. Specifically, we design a set of ablations where we only utilize the text part of the visual instruction tuning data to tune the LLMs, and draw a comparison with the models tuned with both the visual inputs and the corresponding texts.

As shown in fig. 2, we observe that models with text-only visual instruction tuning can largely attain comparable alignment performance with the vanilla visual instruction tuning baseline where both images and texts are used. While additionally including visual inputs yields seemingly "modest" alignment improvements,

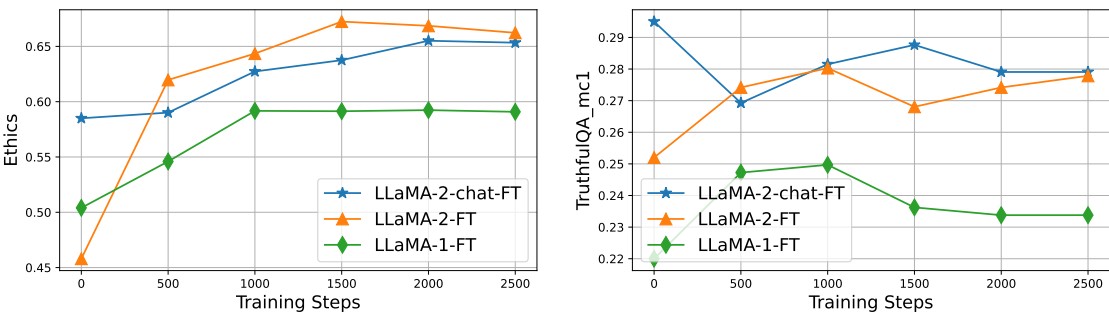

Figure 4: Results of three MLLMs on `Ethics` (*left*) and `TruthfulQA-mc1` (*right*) during MLLM visual instruction tuning.

we stress that these gains are consistent across different LLMs, tuning methods, and alignment tasks. For example, this can be verified across three model variants, resulting in an average accuracy improvement of 2.5% across three sub-tasks presented in fig. 2.

This finding supports the idea that visual instruction tuning can improve model performance, even when trained on GPT-4-generated data. Notably, these two techniques—visual instruction tuning and GPT-4 data training—can be used simultaneously. Additionally, we conduct ablation experiments with text-only data in the later section to better demonstrate the efficacy of visual instruction tuning.

This observation leads to our hypothesis that there exists a promising avenue in leveraging visual data to construct enhanced instruction-tuning datasets. Although textual information plays a significant role in alignment, it is crucial to recognize that this text is inherently grounded in its corresponding real-world visual content; therefore, utilizing such paired information is integral to ensuring strong alignment in LLMs. These findings underscore the multifaceted benefits of visual data: it not only enhances alignment quality but also contributes significantly to the creation of more accurate instruction-tuning datasets.

**Types of Visual Instruction Data Matters.** We further extend our investigation to understand how varying types of visual instruction-tuning data affect LLM alignment. Specifically, we utilize data from LLaVA (Liu et al., 2023b), which categorizes visual instruction tuning data into three groups: `Conversation`, `Details`, and `Reasoning`. Each group comprises 20k data points, sampled from the original training splits. For a fair comparison, we also take a uniform sample of 20k from the full 80k visual instructions to form the baseline group. We tune LLaMA2 and LLaMA2-chat with each data group (of 20k data points) separately, and report the results in fig. 3.

Our analysis reveals that, in general, conversational data has a greater impact on improving LLMs' performance on the `Ethics` task, resulting in an improvement of ~15% on MM-LLaMA2-ft and ~3% on MM-LLaMA2-chat-ft. Conversely, reasoning and details data tend to be more effective in improving performance on `TruthfulQA`, yielding gains of more than 2% and 6% on these two models. This suggests that a targeted approach, leveraging the unique strengths of each data type, can facilitate more nuanced and effective instruction tuning for LLM alignment.

**Imperfect Match Between Multi-Modal and NLP Objectives.** The incorporation of visual information has shown to benefit the ethical and truthful aspects of LLMs. In fig. 4, we present the model performance on the `Ethics` and `TruthfulQA` tasks during the multi-modal LLM finetuning stage. At the initial stage of visual instruction tuning, there is a noticeable improvement in most models for both aspects within the first 1000 training steps. Specifically, the scores on `Ethics` task continue to increase as more visual knowledge is incorporated, indicating a well-aligned training objective between visual instruction training and ethical awareness. However, the alignment between incorporating visual perception into LLMs and enhancing model truthfulness may not be optimal, as the scores for truthfulness degenerate with more training steps considered (*e.g.*, two LLaMA2 models achieve their highest `TruthfulQA-mc1` scores at the 1000*th* training step). This finding is consistent with our previous observation that the model's awareness of truthfulness shows less improvement compared to its ethical alignment.

| Models | MMLU (Acc.) | GSM8K (Acc.) | MathQA (Acc.) | sQuAD (F1.) | BoolQ (Acc.) |
|---|---|---|---|---|---|
| LLaMA | 36.8% | 8.0% | 27.7% | 19.5% | 75.1% |
| MM-ft | 27.7% (-9.0%) | 0.9% (-7.1%) | 28.5% (+0.8%) | 9.1% (-10.4%) | 47.5% (-27.6%) |
| Vicuna | 47.2% | 10.0% | 29.0% | 19.3% | 78.1% |
| MM-ft | 44.0% (-3.2%) | 5.4% (-4.6%) | 29.4% (+0.4%) | 10.1% (-9.2%) | 52.5% (-25.6%) |
| LLaMA-3B | 26.7% | 2.4% | 26.4% | 20.7% | 65.6% |
| MM-ft | 26.5% (-0.2%) | 1.7% (-0.6%) | 25.8% (-0.6%) | 8.6% (-12.1%) | 53.6% (-12.0%) |
| MM-lora | 26.8% (+0.1%) | 3.1% (+0.7%) | 26.3% (-0.1%) | 18.8% (-1.9%) | 66.3% (+0.7%) |
| Alpaca-3B | 24.9% | 0.1% | 24.6% | 28.2% | 71.1% |
| MM-ft | 24.5% (-0.4%) | 0.0% (-0.1%) | 25.6% (+1.0%) | 12.1% (-16.1%) | 69.0% (-2.1%) |
| MM-lora | 24.3% (-0.6%) | 0.1% (+0.0%) | 25.4% (+0.8%) | 24.2% (-4.1%) | 71.1% (+0.0%) |
| LLaMA2 | 45.9% | 13.7% | 30.1% | 26.3% | 77.7% |
| MM-ft | 39.4% (-6.5%) | 5.5% (-8.2%) | 29.6% (-0.5%) | 8.5% (-17.8%) | 56.3% (-21.4%) |
| MM-lora | 46.6% (+0.7%) | 15.0% (+1.3%) | 30.4% (+0.3%) | 20.1% (-6.2%) | 77.6% (-0.1%) |
| LLaMA2-chat | 45.8% | 18.2% | 31.1% | 20.1% | 80.7% |
| MM-ft | 45.2% (-0.6%) | 6.2% (-12.0%) | 30.0% (-1.1%) | 10.2% (-9.3%) | 67.0% (-13.7%) |
| MM-lora | 45.9% (+0.2%) | 17.1% (-1.1%) | 30.8% (-0.3%) | 25.5% (+5.4%) | 81.5% (+0.8%) |
| Vicuna-v1.5-7B | 50.0% | 18.0% | 30.0% | 16.9% | 82.1% |
| MM-ft | 50.6% (+0.5%) | 17.4% (-0.6%) | 29.2% (-0.8%) | 18.2% (+1.3%) | 78.3% (-3.8%) |
| MM-lora | 50.4% (+0.4%) | 16.5% (-1.5%) | 29.4% (-0.6%) | 12.5% (-4.4%) | 82.8% (+0.7%) |
| Vicuna-v1.5-13B | 55.8% | 31.9% | 33.8% | 16.3% | 86.2% |
| MM-ft | 56.0% (+0.2%) | 27.5% (-4.4%) | 33.0% (-0.8%) | 18.6% (+2.3%) | 85.1% (-1.1%) |
| MM-lora | 55.7% (-0.1%) | 26.1% (-5.8%) | 32.5% (-1.3%) | 11.8% (-4.5%) | 86.7% (+0.5%) |
| LLaMA3-8B | 65.7% | 75.7% | 42.2% | 42.6% | 86.7% |
| MM-ft | 56.7% (-9.0%) | 71.2% (-4.5%) | 37.6% (-4.6%) | 14.4% (-28.2%) | 71.9% (-14.3%) |
| MM-lora | 64.7% (-1.0%) | 72.2% (-3.5%) | 42.5% (-0.8%) | 35.1% (-7.5%) | 87.5% (+0.8%) |
| LLaMA3.1 | 68.2% | 77.7% | 43.9% | 38.4% | 87.2% |
| LLaMA3.2* | 68.0% (-0.2%) | 76.9% (-0.8%) | 43.6% (-0.2%) | 38.3% (-0.1%) | 86.8% (-0.4%) |

Table 3: Performances of both the vanilla LLMs and visual-instruction-tuned LLMs on five NLP capabilities benchmarks. Note that LLaMA3.2* denotes a model with visual capabilities built on LLaMA3.1.

By analyzing the trajectory of model performance on these tasks, we observe that the optimization goals between multi-modal ability and the improved truthfulness and ethics are not perfectly aligned. Though LLMs trained on full visual tuning steps surpass the vanilla LLMs on ethics and truthfulness, the same training budgets designed for multi-modal tasks might not be optimal for models' NLP abilities.

## 3.2 Standard NLP Abilities

Given these LLMs are further fine-tuned with multi-modal data, it might be intuitively expected that their standard NLP capabilities could degrade. Such a phenomenon is commonly referred to as catastrophic forgetting (Kirkpatrick et al., 2017) or in the AI alignment community — the alignment tax (Christiano, 2019; Jensen et al., 2023).

Interestingly, contrary to these assumptions, our results presented in table 3 show that MM-lora (marked in the gray background) results in only an average 0.17% performance decrease across five NLP capability benchmarks and four models, after applying visual instruction tuning. More notably, in certain instances, MM-lora even modestly improves performance on these benchmarks. However, the visually-tuned LLaMA3 series shows an average score drop of 4.5% across these tasks while maintain high scores on Ethics and TruthfulQA in Table 1. Notably, LLaMA3.2 — a visually-enhanced LLM tuned by Meta (Dubey et al., 2024) — demonstrates highly consistent performance with our trained models on Ethics, TruthfulQA, and other NLP benchmarks, further supporting our claim that visual instruction tuning enhances LLMs in ethics and truthfulness.

| Models | Ethics | TruthfulQA-gen | TruthfulQA-mc1 | TruthfulQA-mc2 |
|---|---|---|---|---|
| LLaMA2 | 45.8% | 32.3% | 25.2% | 38.9% |
| Alpaca | 52.1% (+6.3%) | 29.6% (-2.7%) | 27.3% (+2.1%) | 41.6% (+2.7%) |
| text-VI | 65.1% (+19.3%) | 31.2% (-1.1%) | 27.7% (+2.5%) | 39.5% (+0.6%) |
| VI | 65.4% (+19.6%) | 31.5% (-0.9%) | 27.8% (+2.6%) | 40.2% (+1.3%) |
| Orca | 62.9% (+17.1%) | 41.6% (+9.3%) | 33.8% (+8.6%) | 49.3% (+10.4%) |

Table 4: Results of LLaMA2 finetuned on Alpaca data, text-only visual instruction data (text-VI), visual instruction tuning data (VI), and Orca data.

Additionally, MLLMs with a more advanced or larger LLM component tend to perform better on NLP benchmarks. For instance, the multi-modal-tuned Vicuna-v1.5-13B outperforms its 7B variant and the v1.3 counterpart by an average of 5.3% and 15.8% across five NLP tasks, respectively.

In conjunction with the insights from Section 3.1, these observations altogether highlight the ability of visual-instruction-tuned LLMs in both maintaining the strong capability on standard NLP benchmarks and aligning better with human values, not to mention the additional capability of recognizing visual inputs. Such findings pave new avenues for both academic exploration and practical implementations within multi-modal domains. We believe these insights should catalyze further investigations into the tuning of LLMs with multi-modal interactions.

## 3.3 Tuning on Different Vision-Language Data

To better explain the benefit of visual text tuning in the process, we present the results of LLaMA2 finetuned on Alpaca data (Taori et al., 2023), text-only visual instruction data (text-VI), visual instruction data (VI), and Orca data (Lian et al., 2023) in Table 4. To keep the fair comparison, we randomly sample 80K data from Alpaca and Orca data respectively for the training.

We can observe that the visual instruction tuning 1) surpasses Alpaca and text-VI data tuned models in most cases, *i.e.*, average 6.7% and 2.8% improvements over Alpaca on two benchmarks, and 0.2%, -1.5% over text-VI; 2) but lags behind the Orca-tuned model by 1.1% and 16.3% on Ethics and TruthfulQA benchmarks. Considering that the Orca dataset includes Chain-of-Thought (CoT) and complex, nuanced instruction-following data from a diverse array of tasks within the FLAN collection (Longpre et al., 2023), the observed performance gain is reasonable. The CoT reasoning and complex instruction-following examples in Orca offer richer contextual understanding and problem-solving patterns. This observation indicates the insight that upgrading text quality within the visual instruction data could further enhance the model's ethical and truthful reasoning, opening a promising avenue for refining these abilities.

## 3.4 Analysis on Multi-Modal Benchmarks

We hereby test the visual-instruction tuned models on recent multi-modal benchmarks, where five tasks are deployed: `Unicorn` benchmark (Tu et al., 2023a) dedicates evaluating the MLLM ability in safety scenarios, we take two `OODCV-VQA` tasks and `Sketchy-VQA` tasks for testing whether models can well handle OOD visual/text input and sketch images, respectively. `MME` (Fu et al., 2023) consists of two evaluation aspects, *i.e.*, cognition (`CS`) and perception (`PS`) with total 14 VQA tasks;[1] `MSCOCO` (Lin et al., 2014) and `Flickr30k` (Young et al., 2014) captioning tasks are commonly used benchmarks in the field of image caption generation. We report the zero-shot CIDEr (Vedantam et al., 2015) scores (with three text-only QA examples) on the test set from the Karpathy split (Karpathy & Fei-Fei, 2015). `POPE` (Li et al., 2023c) is used to evaluate the level of object hallucinations in MLLMs, which consists of three versions of balanced yes/no VQA tasks considering objects in the given image. It is built upon `MSCOCO`-2017 dataset (Lin et al., 2014). Additionally, We also make use of the image corruptions proposed in `ImageNet-C` (Hendrycks & Dietterich, 2019) to measure the performance of the MLLMs on corrupted images for `MSCOCO` task (denoted as `MSCOCO-C`).[2]

---

[1]We exclude `landmark` and `artwork` tasks to accelerate the evaluation process.

[2]For corrupted images, we report the average results of tested models on four noises (gaussian noise, defocus blur, contrast, brightness) across three severity levels (1, 3, 5)

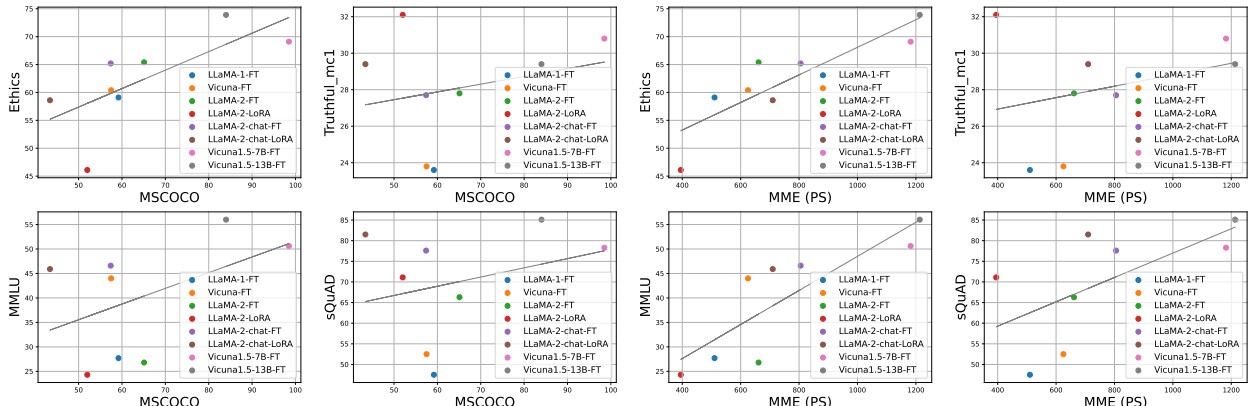

Figure 5: Results of eight different MLLMs on two multi-modal and four NLP tasks.

| Models | Unicorn oodcv / sketch | MME$^{CS}$ | MME$^{PS}$ | COCO | Flickr30k | POPE R / A / P |
|---|---|---|---|---|---|---|
| MM-LLaMA-ft | 45.2 / 80.9 | 199.3 | 510.5 | 59.2 | 27.1 | 65.7 / 57.8 / 59.9 |
| MM-Vicuna-ft | 58.4 / 82.7 | 270.7 | 625.2 | 57.5 | 24.6 | 76.5 / 66.5 / 73.8 |
| MM-LLaMA2-ft | 55.0 / 83.6 | 237.1 | 661.3 | 65.1 | 31.6 | 65.0 / 55.4 / 56.3 |
| MM-LLaMA2-lora | 52.3 / 79.6 | 200.0 | 395.0 | 52.0 | 26.2 | 50.8 / 50.4 / 50.6 |
| MM-LLaMA2-chat-ft | 54.5 / 80.9 | 234.6 | 805.4 | 57.4 | 26.7 | 69.8 / 57.9 / 60.3 |
| MM-LLaMA2-chat-lora | 53.1 / 82.8 | 228.6 | 709.8 | 43.4 | 23.0 | 65.9 / 56.8 / 59.2 |
| MM-Vicuna-v1.5-7B | 58.4 / 80.4 | **320.4** | 1182.0 | **98.5** | **62.8** | **89.3 / 79.7/ 85.5** |
| MM-Vicuna-v1.5-13B | **59.7 / 87.8** | 287.9 | **1213.3** | 84.0 | 51.3 | 87.4 / 78.7 / 84.1 |

Table 5: Performances of our MLLM family on five widely employed multi-modal benchmarks. We test models on oodcv and sketch sub-tasks in the `Unicorn` benchmark (Tu et al., 2023a) and Random (R), Adversarial (A), and Popular (P) in `POPE` (Li et al., 2023c).

**Enhanced MLLMs Expand NLP Capabilities.** *Is a better visual reasoner also a better NLP task solver?* In fig. 5, we illustrate the correlation between model performance in multi-modal and NLP tasks. The results reveal positive correlations across tasks from different domains, suggesting that the visual reasoning abilities of the eight models analyzed contribute to their improved performance on NLP benchmarks. Notably, the average coefficient of determination score across these eight scenarios is 0.482, indicating a moderate correlation. In specific cases, such as pairing `MME (PS)` with `Ethics` and `MMLU`, both of the coefficient scores exceed 0.72, indicating a very strong correlation between these tasks. This evident correlation explains our finding of stronger MLLMs can lead to expanded NLP capacities. But unlike the misaligned objectives observed between MLLM multi-modal and NLP abilities during training in Sec. 3.1, this finding might not be surprising, as MLLMs rely heavily on language to reasoning and expression. It is plausible that an improved LLM (*i.e.* better training data, larger model scale) could enhance the expressive abilities of an MLLM, resulting in a close correlation between abilities across different modalities.

***Aligned* LLMs *vs. Unaligned* LLMs on Multi-Modal Benchmarks.** In table 5, MLLMs incorporating *aligned* LLMs (*e.g.*, Vicuna, LLaMA2-chat) have demonstrated top performance in comprehensive and challenging tasks such as `Unicorn`, `MME`, and `POPE`. Specifically, MM-Vicuna-ft and MM-LLaMA-chat-ft outperform their corresponding vanilla MLLM counterparts by an average of 164.9 on `MME` and 7.5% on `POPE`. Despite the incorporation of text-aligned LLMs, MLLMs still exhibit unexpected shortcomings when compared to models that use vanilla LLMs, particularly when evaluated on three traditional vision-text tasks. For instance, MLLMs show an average drop of 4.2 CIDEr across two captioning tasks. This could be attributed to the nature of the captioning task itself, which is more generation-focused and less about following instructions as seen in more challenging QA tasks. As a result, MLLMs that use unaligned LLMs may perform better on captioning tasks, as they are less constrained by alignment requirements and are better suited for the generation-centric nature of captioning.

| Models | COCO (CIDEr) | COCO-C (CIDEr) |
|---|---|---|
| MM-LLaMA-ft | 59.2 | 48.6 (-17.9%) |
| MM-Vicuna-ft | 57.5 | 46.0 (-20.0%) |
| MM-LLaMA2-ft | **65.1** | **54.6** (-16.1%) |
| MM-LLaMA2-lora | 52.0 | 43.2 (-16.9%) |
| MM-LLaMA2-chat-ft | 57.4 | 47.5 (-17.2%) |
| MM-LLaMA2-chat-lora | 43.4 | 33.8 (-22.1%) |

Table 6: Performances of the MLLM family on MSCOCO (Lin et al., 2014) with corrupted visual inputs.

**Need for Studying Multi-Modal Alignments.** Although text-aligned models like Vicuna and LLaMA2-chat have proven effective, their MLLM variants perform poorly on corrupted images, as shown in table 6. Not only do these models underperform compared to MLLMs that do not use instruction-tuned LLMs, but they also exhibit a performance drop of over 17% when evaluated on corrupted images, compared to clean ones. This decline is even greater than the drops observed in MM-LLaMA-ft and MM-LLaMA2-ft. This suggests that while visual instruction tuning enhances the truthfulness and ethical behavior of LLMs in the language domain, these MLLMs still face their unique challenges in the multi-modal context, especially when dealing with corrupted or imperfect visual inputs (Tu et al., 2023a; Dong et al., 2023).

## 4 Related Work

**Alignments.** The alignment of AI systems to human values is an important topic for today's advanced AI systems, from testing model robustness to out-of-distribution shifts (Hendrycks & Dietterich, 2019; Hendrycks et al., 2021a; Zhao et al., 2022) to adversarial attacks (Hendrycks et al., 2021b; Eykholt et al., 2018; Xie et al., 2020), many works have been proposed. The recent development of LLMs has revolutionized natural language processing and has been widely adopted in various applications. Thus, concerns regarding the honesty and truthfulness of these models have also emerged, prompting alignment researchers to investigate the ethical implications and potential risks associated with their deployment. TruthfulQA (Lin et al., 2022) is proposed to measure how LLMs imitate human misconceptions. And Ethics (Hendrycks et al., 2020) is used to assess a language model's knowledge of basic concepts of morality.

Advanced techniques for aligning language models with human preference are also popular these days, from RLHF (Ouyang et al., 2022) to DPO (Rafailov et al., 2023) optimization, the alignment training paradigms have shifted fast recently. Reinforcement learning as one of the most popular solutions to enhance the truthful and ethical awareness of LLMs have long been discussed. Recent works explored various approaches for such purpose, including progressive rewarding (Gao et al., 2024), optimized RL reward function (Bai et al., 2022), and applications as well as benchmarking (Zhang et al., 2024; Li et al., 2024). The concept of LLM alignment has also gradually switched from human-supervised (Ouyang et al., 2022) to the paradigm of incorporating other AI model supervisions (Lee et al., 2023), and to most recently the employment of weak signals (Burns et al., 2023). Given the popularity of the use of large language models, adversarial attacks on LLMs have also been explored (Zou et al., 2023). In this work, we present our findings on how visual instruction tuning can help the LLMs align with human values, our results show impressive performance boost on these datasets without explicit prompting such behaviors.

**Multi-Modal and Large Language Models.** In light of the rapid evolvement of large language models (LLMs), recent studies about multi-modal systems have turned their focuses from incorporating fine-grained multi-modal data (Liang et al., 2021; Tu et al., 2023b) to integrating powerful LLMs with few-shot capability. More recently, some instruction-tuned MLLMs have emerged, showing excellent generalization ability in unseen VL tasks (Zhu et al., 2023; Liu et al., 2023b; Ye et al., 2023; Li et al., 2023a; Dai et al., 2023). For example, MiniGPT4 (Zhu et al., 2023) is built upon QFormer (Li et al., 2023a) and Vicuna (Zheng et al., 2023) and only activates the linear layer connecting the vision encoder and LLM. LLaVA (Liu et al., 2023b;a) projects the output of a vision encoder to word tokens and trains both the VL connector and the LLM on synthetic data. mPLUG-owl (Ye et al., 2023) tunes LLaMA with a query-based VL connector using both text-only and vision-language instruction data. InstructBLIP (Dai et al., 2023) uses BLIP2 (Li et al., 2023a)

as the backbone but is additionally instruction-tuned on a collection of VL datasets. Other multi-modal LLMs in a vast range of modalities sparkles insights and deployment in real-world applications (Zhang et al., 2023c; Bai et al., 2023; Zhang et al., 2023a; Liu et al., 2024).

Despite the rapid growth in this domain, recent benchmark works have shown that current multi-modal large language models still suffer from problems like being unable to handle counterfactual statements (Zhang et al., 2023b; Yu et al., 2023; Tu et al., 2023a; Lee et al., 2024), hallucination (Li et al., 2023b; Zhou et al., 2023), and simple answer set permutations (Zong et al., 2023). In our work, we demonstrate a new perspective on these MLLMs – tuning LLMs with multi-modal data greatly helps align them with human values.

## 5 Discussion, Conclusion, and Future Work

**More Aligned Objectives between Multi-Modal and NLP Abilities.** Our exploration shows that training on multi-modal instruction tuning data can also benefit the LLMs' factual accuracy and ethics. In fig. 4, we have shown that these alignment-focused metrics improve while training proceeds on multi-modal. However, current multi-modal data is not designed for alignment, the main focus is still eliciting language models with multi-modal perception. Our results demonstrate a promising new avenue for developing models to understand and interact with the world more truthfully, and also suggest the need for exploration to identify appropriate tasks that can effectively improve these two aspects simultaneously. And we hope our paper could inspire discussions on this direction.

**Exploring the Training Framework.** In our pilot study, we have shown the results of using image-based instruction fine-tuning data to support our findings that leveraging multi-modal interactions could yield more aligned models. Based on our results, it is reasonable to assume that introducing multi-modal data to the pre-training stage could also yield more aligned models. For example, the Gemini family models could be an interesting case for study (Gemini Team, 2023). Understanding how to instruction fine-tune the base model for multi-modal ability and alignment is another direction worthy of exploration. Our study explores full parameter fine-tuning as well as LoRA parameter efficient fine-tuning. It can be beneficial to study how varies types of parameter-efficient fine-tuning techniques helps (Kopiczko et al., 2024; Zhao et al., 2024). Besides the training techniques, the training data can also be explored, how should we create a mixture of data for fine-tuning, how to determine the ratio of multi-modal data to text-only data (Ye et al., 2023; Liu et al., 2023a), and how to extend to other modalities other than images. These exploration could help gives a more practical and comprehensive guide.

**Conclusion.** In this study, we offer preliminary findings that underscore the potential of enhancing the truthfulness and ethical alignment of LLMs through visual instruction tuning. Remarkably, even without prompts tailored for truthfulness or ethical behaviors, our approach to tuning LLM weights using visual instruction datasets yielded significant improvements in both the `TruthfulQA` and `Ethics` benchmarks. Notably, such improvements are even stronger than that of RLHF, which tunes LLMs with a huge corpus of human-aligned data points. The follow-up analysis demonstrates the importance of instruction data quality for improving aligned values in MLLMs, as well as specific types of data models employed for applying to different alignment tasks.

**Future Work.** In light of our findings, we advocate for future research endeavors to focus on devising innovative methodologies for crafting visual instruction data that can more effectively align LLMs. Exploring novel MLLM architectures could also be a fruitful avenue. We hope fostering LLM interactions with real-world environments may emerge as a pivotal strategy for achieving superior model alignment.

## Acknowledge

This work is partially supported by a gift from Open Philanthropy. We would like to thank Center for AI Safety, Microsoft Accelerate Foundation Models Research Program, and the OpenAI Researcher Access Program for supporting our computing and resource needs.

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
