# OpenReview forum: "Sight Beyond Text: Multi-Modal Training Enhances LLMs in Truthfulness and Ethics"
_TMLR — Accepted by TMLR_

### Review · Reviewer_NBFb · 2024-10-05

**Summary Of Contributions:**

This paper primarily investigates the benefits of incorporating visual instruction tuning data to enhance the truthfulness and ethical alignment of large language models (LLMs). To support this claim, the authors conduct extensive experiments: i) Applying visual instruction tuning to various LLMs of different types and sizes. ii) Comparing the performance of models fine-tuned on pure text data versus those fine-tuned on visual instruction tuning data. iii) Comparing the performance of GPT-4-turbo with GPT-4V.

**Audience:**

Yes

**Claims And Evidence:**

Yes

**Requested Changes:**

See weakness above

**Strengths And Weaknesses:**

### Strengths

1. The point studied in this paper is novel and significantly important for understanding the influence of visual instruction tuning on large language models (LLMs). Most existing works indicate that LLMs undergoing multimodal training often exhibit degenerate performance in NLP tasks.
2. The authors design extensive experiments to support the claims made in this paper.
3. In addition to the main results—demonstrating that visual instruction tuning can improve truthfulness and ethical alignment—the authors also present other experimental findings, such as the correlation between strong multimodal LLMs (MLLMs) and strong LLMs.
4. This paper is well-written and easy to read.

### Weaknesses

The experimental results cannot fully support the claim that visual instruction tuning can improve truthfulness and ethical alignment:

1. The experiment settings in Figure 2 are not comparable to each other. Naively extracting the text part of the visual instruction tuning data makes the instruction tuning data incomplete, severely degrading its quality.
2. The gap between GPT-4-turbo and GPT-4V is too small, making the difference insignificant.
3. Instruction tuning with the LLaVA dataset cannot fully support the claim made in this paper:

    3.1 The LLaVA instruction tuning dataset is more like a pure-text instruction tuning dataset, as it is constructed using text-only GPT-4.

    3.2 The study lacks other visual instruction tuning datasets. The authors are highly encouraged to incorporate additional visual instruction tuning datasets, such as Cambrian-10M[1] or ALLaVA[2], to make the claim more general and valid.

4. Complementing point 1 above, the authors are also encouraged to compare models fine-tuned with visual instruction tuning datasets and pure-text instruction tuning datasets. These datasets should be of similar size, and the base LLM should use a version that has not been fine-tuned.

[1] Cambrian-1: A Fully Open, Vision-Centric Exploration of Multimodal LLMs

[2] ALLaVA: Harnessing GPT4V-Synthesized Data for Lite Vision-Language Models

---

> ### Author Response · Authors · 2024-11-03
>
> 1. **Explain Figure 2**: Thank you for the insightful question. The experiment in Figure 2 aims to explore the impact of visual data in visual instruction tuning. Specifically, we seek to understand *how different modalities (image and/or text) within visual instruction data contribute to LLM alignment*. To do this, we used only the text portion of the visual instruction data to tune the LLMs, allowing us to compare with fully visually-tuned models. Regarding potential effects on LLM performance, our results show that text-only visual instruction also enhances model performance on Ethics and Truthfulness. We also incorporate other text-only tuning results in Table 4.
> 2. **Add more GPT-4 models and LLaMA-3 series**: Thank you for the suggestion. In the revised submission, we incorporated your advice and used the more recent GPT-4 Turbo and GPT-4V Turbo (1201) as well as five more LLaMA-3 series models on Ethics and TruthfulQA tasks. For detailed analysis and results, please refer to the revised submission and general response.
> 3. **Add text-only instruction tuning data**: Thank you for your suggestions in adding more tuning data. We incorporated your advice and added Alpaca and Orca text instruction data (downsampled to the same 158K size) for LLM finetuning. Please refer to the general response and the revised manuscript for more analysis and details.

---

### Review · Reviewer_NThr · 2024-10-13

**Summary Of Contributions:**

The paper explores how multi-modal large language models (MLLMs) trained on both text and visual data can significantly improve their performance in natural language processing tasks related to truthfulness and ethical alignment. The paper claims that visual instruction-tuned models improve truthfulness and ethics compared to their purely text-tuned counterparts. Also,
models that leverage visual data during fine-tuning generally perform better, even when tested without visual input. Moreover, visual instruction tuning enhances the alignment of LLMs with human values, suggesting the potential for more aligned AI systems. The paper encourages further research into integrating visual data to advance the alignment of LLMs. It suggests that broadening the type of data used in LLM training could be pivotal in AI alignment.

**Audience:**

Yes

**Claims And Evidence:**

Yes

**Requested Changes:**

1. Llama3 was released earlier this year. Given your claims that "stronger base LLMs are more capable during the multi-modal tuning process" and "the ability of visual-instruction-tuned LLMs in both maintaining the strong capability on standard NLP benchmarks and aligning better with human values", testing Llama3-8B would provide strong validation of your approach.
2. In Table 2, you compare GPT-4 and GPT-4V. To strengthen the evaluation, please also consider comparing Llama3/3.1 with Llama3.2.
3. The improvement of GPT-4V over GPT-4 on Ethics is 0.4%, which is marginal. Do you have variance data or significance tests to assess the robustness of this improvement?
4. In training, you tune connectors for each LLM with 595k image-text pairings. Since connectors are integral to the tuning process, why aren't those parings counted as part of the training data? Although they are not instructions-following data, they involve the entire tuning process. With that amount of tunning data, you still use less data than RLHF.
5. In Figure 2, should the blue text represent "text-only instructions"?
6. The text in the figure legends for Figures 3 and 4 is too small and should be adjusted for clarity.
7. Please provide relevance scores in either Figure 5 or the main text for a more comprehensive analysis.

**Strengths And Weaknesses:**

Strengths:
1. The paper introduces an innovative perspective by showing that multi-modal training improves multi-modal tasks and enhances ethical and truthful behavior in NLP tasks. This opens up new directions for AI alignment research.
2. The experiments demonstrate that even with fewer human annotations, models fine-tuned with visual instruction outperform heavily human-annotated models in truthfulness and ethics benchmarks.
3. The paper evaluates multiple models and compares text-only and multi-modal models using well-established benchmarks like TruthfulQA and Ethics, providing strong support for the conclusions.
4. The paper benefits from visual-text synergies in improving alignment and suggests a new paradigm for future LLM development.

Weakness:
1. While the paper demonstrates improved performance, it lacks a deeper analysis of why visual instruction tuning helps with truthfulness and ethics.
2. The paper doesn't address instances where visual instruction tuning may not be effective. Negative results could be discussed more thoroughly to provide a balanced view.
3. Although Figure 5 shows the relevance improvements between multimodal and NLP tasks, the variance is significant, making the results less confident.
4. The paper briefly mentions the potential trade-offs between multi-modal tuning and standard NLP abilities. The finding is intriguing. However, it does not provide further analysis or relevant experiments.

---

> ### Author Response · Authors · 2024-11-03
>
> 1. **Deeper Analysis**: Thank you for your suggestions. We have added additional explanations in Sections 1 and 2. Our main claim is that current multi-modal LLMs handle visual tokens by learning to bridge the gap between visual and language modalities. While language tokens capture many real-world scenes, visual information provides a richer, more nuanced context that connects closely to "factual knowledge" from everyday life, particularly in areas related to truthfulness and ethics. Thus, training LLMs to learn and process visual tokens enhances their performance in aspects like ethics and truthfulness.
> 2. **Discussions about negative results**: In Section 3.1, we discuss the imperfect alignment between multi-modal and NLP objectives. Our experiments primarily focus on evaluating improvements in ethics and truthfulness through language interactions between models and humans (i.e., question-answering or text generation tasks). This focus can sometimes lead to fluctuations in performance regarding ethical and truthful standards. However, the model consistently performs well in these areas compared to other NLP tasks, reinforcing our claim that visual instruction tuning supports ethical and truthful knowledge acquisition. Additionally, while the previous Vicuna-v1 model with visual tuning significantly lagged behind its vanilla version, our expanded experiments with more recent models—such as Vicuna-v1.5 and LLaMA-3/3.1—again confirm that visual instruction tuning enhances LLMs in ethics and truthfulness. The underperformance of Vicuna-v1 may stem from its less aligned instruction training, which does not fully reflect human intentions.
> 3. **Variance in Figure 5**: Thank you for asking the question. We followed your suggestion and did experiments to obtain the coefficient of determination scores between every task pair in Figure 5. The average coefficient of determination score across these eight scenarios is 0.482, indicating a moderate correlation. In specific cases, such as pairing MME (PS) with Ethics and MMLU, both of the coefficient scores exceed 0.72, indicating a very strong correlation between these tasks. This evident correlation explains our finding of stronger MLLMs can lead to expanded NLP capacities.
> 4. **NLP ability and multi-modal trade-off**: Thank you for your question. Since LLMs are initially trained for NLP tasks, gaining new multi-modal abilities can sometimes impair their NLP performance (as also mentioned by Reviewer NBFb). This observation aligns with the concept of "catastrophic forgetting," where visual instruction tuning may cause MLLMs to prioritize multi-modal content, potentially diminishing some language capabilities. However, as shown in Figure 5, there remains a strong correlation between the model's existing NLP abilities and its newly acquired multi-modal skills. This outcome is perhaps unsurprising, as MLLMs rely heavily on language for reasoning and expression. It’s reasonable to expect that an improved LLM (e.g., with better training data or a larger model scale) could enhance an MLLM’s expressive capabilities across modalities, resulting in a close correlation between skills.
> 5. **Add Models**: Thank you for your suggestions in adding more models. For closed-API models, we add results of two more recent OpenAI GPT models: GPT-4 Turbo (1205) and GPT-4V Turbo (1205). For open-weight models, we add LLaMA-3 series with 5 more models: LLaMA-3-8B, MM-LLaMA-3-8b-ft,  MM-LLaMA-3-8b-lora, LLaMA-3.1, and LLaMA-3.2. For detailed results and analysis, please refer to the revised submission and general response.
> 6. **Training data details**: In our experiments, we mainly focus on the performance gain of LLMs. At the first stage of MLLM training, we only activate the connector between the vision encoder and the LLM on 595k image-text pairs and leave the LLM untouched. That’s why we only count the second stage training into the LLM training, where we activate the LLM in the model. We have revised the description about the training stage to avoid confusion in this manuscript.
>  Apologize for confusion made in the previous version of training data.
> 7. **Suggestions about the paper presentation**: Thank you for your meticulous review and suggestions on the paper's presentation. In Figure 2, "text-only visual instruction" refers to training LLMs using visual instruction data without images. The intent is to highlight that this data shares the same textual content as in visual instruction tuning. Additionally, we have revised other formatting issues according to your guidance.

---

### Review · Reviewer_ah3g · 2024-10-20

**Summary Of Contributions:**

This paper explores the effect of fine-tuning of LLMs with visual instruction tuning data. The authors use two text-only benchmarks, namely Ethics and TruthfulQA, and they find that some models show significant improvements. A range of open weight LLMs are compared, covering a range of 3B to 13B parameters, and different data. The fine-tuned models are further evaluated on language benchmarks and multi-modal benchmarks.

**Audience:**

Yes

**Broader Impact Concerns:**

I don't think that this work requires a Broader Impact Statement section.

**Claims And Evidence:**

No

**Requested Changes:**

1) Design experiments that can answer the question whether the observed differences in alignment/truthfulness can be explained by visual instruction tuning, or by fine-tuning on GPT-4 generated data.

2) Develop a hypothesis why visual instruction tuning would improve on the ETHICS benchmark. Figure 1 gives some intuition why TruthfulQA would be affected positively by visual instruction tuning data (although also for TruthfulQA, it would be interesting to discuss this in more detail, with a focus on the different categories of the benchmark questions).

3) The paper claims "visual instruction tuning is able to yield empirical advantages that surpasss those of RLHF" – in the context of this quote (and in the related works section), I would like to see some references that examine the effect of RLHF (or LLM alignment in general) on the presented benchmarks (ETHICS / TruthfulQA). If these two benchmarks are relevant to modern LLMs, then I would expect additional studies to exist.

4) Add baseline numbers to table 1 (see also comment above).

5) Explain in more detail why there is such a large difference between -ft and -lora models.

6) Explain why does Vicuna-7B lose so much performance on alignment/truthfulness after visual instruction tuning? Currently the authors state that "visual instruction tuning is limited at enhancing the alignment of models previously fine-tuned via instruction tuning (e.g. Vicuna)" – but table 1 shows not a limited improvement, but a marked worsening for the "MM-ft" variant of the "Vicuna" model.

7) Add MM-lora entries for all models.

8) Section 3.1 paragraph "Types of Visual Instruction Data Matters" mentions that LLaVA has 80k data points. But [(Liu, 2023b)](http://arxiv.org/abs/2304.08485) mentions 158k data points (77k reasoning, 58k conversation, 23k detailed description) – why the difference?

9) I would like to see some motivation around the model selection: for example, why were Mistral, Gemma, Gemma2, LLaMA3 not included? Why were the model sizes 3B,7B chosen, but 13B was only used with Vicuna (but not with LLaMA and LLaMA2 that also have 13B variants)

10) Include more evaluation details: How exactly was the Ethics benchmark evaluated? [(Hendrycks, 2020)](https://arxiv.org/abs/2008.02275) mentions fine-tuning and fewshot evaluation (in case of fewshot: which prompt was used?). Why was the Rouge metric chosen for TruthfulQA-gen (and not the GPT-judge)? What does -mc1 and -mc2 refer to?

11) Add baseline numbers to table 4.

12) Typo: Table 1 captions says "TruthfullQA-gen" instead of "TruthfulQA-gen"

13) Typo: in Discussion section, the reference should be "(Gemini Team, 2023)" and not "(Team et al, 2023)"

**Strengths And Weaknesses:**

S1) The study examines a variety of open weight LLMs with respect to their alignment (using the ETHICS benchmark) and truthfulness (using the TruthfulQA benchmark) before and after visual instruction tuning. Alignment and truthfulness are important aspects of LLMs, and as these models are often modified to support multimodal input, it is important to also assess the change in performance on these tests after such modifications.

S2) The authors do not only look at alignment/truthfulness, but also evaluate the models on language benchmarks and a set of multi-model benchmarks.

W1) In its current form, I'm not convinced by the paper that the observed change in alignment/truthfulness is due to visual instruction tuning. The authors only use a single dataset that is generated by GPT-4 (Section 3 – [(Liu, 2023b)](http://arxiv.org/abs/2304.08485)), so there is a strong confounder "fine-tuning on data generated by GPT-4". Figure 2 from the paper also suggests that a large part of the observed effect is due to information contained in the text-only part of the LLaVA visual instruction tuning data. Further, the effect is much weaker on models that have already been trained on ShareGPT data (e.g. Vicuna), which points towards the possibility of such a transfer. Note the high scores of GPT-4 reported in Table 2.

W2) I find the main results in Table 1 hard to interpret:

W2.1) In some cases (e.g. Vicuna), there is a large drop in alignment/truthfulness – why?

W2.2) There seems to be a large difference between -ft and -lora (and in opposing directions). Taken together with non-monotonic training curves in Figure 4 (right), and with the opposite effect when adding visual information on some of the metrics reported in Figure 2, this makes me wonder how significant the observed changes are.

W2.3) There are no references anchoring any of the numbers in previous literature.

W2.4) Why is Rouge reported for TruthfulQA-gen? The paper (Appendix B – [(Lin, 2022)](http://arxiv.org/abs/2109.07958)) mentions a GPT-judge with much better correlation to human judgment. What is mc1 and mc2 (I couldn't find a reference to these accuracies in [(Lin, 2022)](http://arxiv.org/abs/2109.07958)).

---

> ### Author Response · Authors · 2024-11-03
>
> 1. **Explain the performance improvement**: Thank you for your question. In the original manuscript, we designed the experiment in Figure 2 to train LLMs using image-free visual instruction data. Our results confirm that visual data enhances LLM performance in ethics and truthfulness compared with instruction data without images. This finding supports the idea that visual instruction tuning can improve model performance, even when trained on GPT-4-generated data. Notably, these two techniques—visual instruction tuning and GPT-4 data training—can be used simultaneously. Additionally, we conducted experiments with text-only data, as shown in Table 4; please refer to the general response for more details.
> 2. **Develop hypothesis**: Thank you for your suggestions. We have added additional explanations in Sections 1 and 2. Our main claim is that current multi-modal LLMs handle visual tokens by learning to bridge the gap between visual and language modalities. While language tokens capture many real-world scenes, visual information provides a richer, more nuanced context that connects closely to "factual knowledge" from everyday life, particularly in areas related to truthfulness and ethics. Thus, training LLMs to learn and process visual tokens enhances their performance in aspects like ethics and truthfulness.
> 3. **Explanations about advantages over existing RLHF**: Thank you for your question. In this paper, we aim to interpret the advantages of visually-tuned models over RLHF-ed models (e.g., LLaMA-chat, Alpaca-3B) by comparing their performance in ethics and truthfulness. As shown in Table 1 and discussed in Section 3.1, our analysis reveals that LLaMA2's performance on the Ethics task improved by 19.6%, surpassing the vanilla LLaMA2-chat and Alpaca-3B by 6.9% and 11.3%, respectively. This analysis supports our claim that the visual instruction tuning has the potential to benefit LLMs more effectively than certain RLHF-ed models. We have also added some references in this area in the revised manuscript.
> 4. **Explain finetuning and LoRA-tuning gap**: Thank you for raising the question. At the second stage of MLLM training, we activate the LLM component along with the connector. While finetuning the LLM part in MLLM with full parameter activation leads to better multi-modal performance, which has also been validated by other works, it generally underperforms LoRA-tuned MLLMs in TruthfulQA task (i.e., 2.2%, 2.4%, and 2.6% improvement of LoRA tuning compared with finetuning for Vicuna-1.5-7B, Vicuna-1.5-13B, LLaMA-3.1, respectively). But LoRA-tuned MLLM lags behind finetuned ones on Ethics by an average of 7.2% across 7 model variants. This observation suggests that the ethics aspect aligns better with the multi-modal objective in visual instruction tuning than the truthfulness aspect as discussed in the subsection of ‘Imperfect Match Between Multi-Modal and NLP Objectives’ later in the paper.
> 5. **Explain negative results**: While the previous Vicuna-v1 model with visual tuning significantly lagged behind its vanilla version, our expanded experiments with more recent models—such as Vicuna-v1.5 and LLaMA-3/3.1—again confirm that visual instruction tuning enhances LLMs in ethics and truthfulness. The underperformance of Vicuna-v1 may stem from its less aligned instruction training, which does not fully reflect human intentions. We thus have moved its results to the appendix.
> 6. **LLaVA tuning data**: Apologize for confusion made in using the LLaVA training data. We realized that the training data description was not clear enough in the last submission. We took the original LLaVA training data which contains 80K unique images and 158K training instances in total. We would like to thank you for pointing out this misclarification and we have corrected this error in the revision. We made sure that the same data were employed across all models to keep them equal competitions.

---

> > ### Author Response · Authors · 2024-11-03
> >
> > 7. **Motivation of model selection**: In the previous version of the manuscript, we selected the LLaMA2 and Vicuna-v1.5 series due to their prevalence as widely used open-weight models, which are foundational for popular MLLMs like the LLaVA series (v1.5, Next). In the revised submission, we expanded our evaluation and analysis by adding several more recent models: GPT-4 (1201), GPT-4V (1201), LLaMA-3, LLaMA-3.1, and LLaMA-3.2. This selection was motivated by the LLaMA-3 series’ standing as the most powerful open-weight LLMs and OpenAI’s GPT-4 series as the leading closed-API models. Please refer to the revised submission and the general response for more details.
> > 8. **More evaluation details**: Thank you for your suggestions in adding details. We mainly employ the open-source and widely-used codebase: lm-evaluation-harness (https://github.com/EleutherAI/lm-evaluation-harness) for LLM evaluation. As in the original TruthfulQA evaluation, the Rouge and BLEU metrics are employed. We evaluate all models using the same metrics, which makes it a fair comparison. As for the other metrics, we refer to the official TruthfulQA repository (https://github.com/sylinrl/TruthfulQA?tab=readme-ov-file#tasks) to name the ‘single-true’ and ‘multi-true’ metrics as mc1 and mc2. We have added more evaluation details in the revised manuscript.
> > 9. **Add model results in Table1, Table3, and Table 4**: Thank you for your suggestions. We have added LoRA results in Table1 and Table3, and we have also filled the missing multi-modal scores in Table4. Please refer to the revised manuscript for detailed results and analysis.
> > 10. **Typos, paper presentation improvement**: Thank you so much for your meticulous review and suggestions. We’ve revised the paper according to your advice.

---

> > > ### Comment · Reviewer_ah3g · 2024-11-07
> > >
> > > I would like to thank the authors for the updates to the manuscript – adding additional models and completing tables with LoRA results fills in some important gaps, as does the added Table 4 with evaluation of LLaMA2 fine-tuned on different text-only data.
> > >
> > > The majority of my requested changes were addressed, but I still find some crucial points not satisfyingly answered:
> > >
> > > 1. I have looked at Figure 2 in the original paper, but I do not find that this figure answers my question whether the difference is explained by visual instruction tuning data or GPT-4 generated data. The figure shows a modest difference in some models/benchmarks (MM-LLaMA2-ft), and even an *increased* effect in some (e.g. MM-LLaMA2-lora/TruthfulQA-mc2). In order to answer the question whether "visual instruction tuning data" is the explanatory characteristic (vs. e.g. GPT-4 generated data), I would like to see an improvement of models fine-tuned on visual-instruction-tuning data that was *not* generated by GPT-4, and compare this to (not visual) instruction-tuning data that was generated by GPT-4. The added Table 4 seems to further suggest that it is the data generated by a more powerful instruction-tuned model, and not the multi-modal component that improves the scores on Ethics and ThruthfulQA: The data from the weakest model (Alpaca, generated from text-davinci-003) improves the least, the difference between text-VI and VI is very small, and the more powerful text-only Orca data works best on TruthfulQA.
> > > 2. I have re-read Sections 1 and 2 in the revised manuscript, but I still don't find a hypothesis, why visual information would help in the *ethics* benchmark. The supposed "factual knowledge" and the "richer real-world details" from visual tokens do not explain why this would improve performance on an ethical benchmark without additional explanation.
> > > 3. From the authors' answer I understand that there is no previous literature about RLHF or similar techniques and the used ethics/truthful benchmarks. This is surprising, and raises the question how relevant these benchmarks really are for modern LLMs. I assume this absence of evaluations from previous work is also the cause why my requested change 4. "Add baseline numbers to table 1 (see also comment above)." was not addressed. Note that Table 5 (formerly Table 4) is also still missing baseline numbers (as requested in my 11.)
> > > 4. As for typos, note that the reference to the Gemini paper in the discussion section now reads "(Team, 2023)" – instead of "(Gemini Team, 2023)".
> > >
> > > After reading the revised manuscript, I still don't think that the data provides sufficient evidence that *multi-modal* training enhances LLMs truthfulness and ethics.

---

> > > > ### Author Response · Authors · 2024-11-09
> > > >
> > > > **GPT-4 generated data**: Thank you for your feedback. We would like to clarify several points regarding our manuscript. In Section 3.1, Effects of Modalities in Visual Instruction-Tuning Data on LLM Alignment, we provide evidence demonstrating that incorporating visual instruction data enhances model performance across two tasks and four different model variants, outperforming text-only data and yielding an average accuracy gain of 1.7% across all four tasks (42.1 vs. 40.4). Notably, **three out of four scenarios** validate that visual cues significantly contribute to improvements in model truthfulness and ethical alignment. While you highlighted the decreased performance observed in the MM-LLaMA2-lora/TruthfulQA-mc2 setting, it is essential to acknowledge that the LLaMA2-chat model, employing the same LoRA tuning approach on TruthfulQA-mc2, showed a substantial 4.6% improvement. Furthermore, an average increase of 1% was observed in other models within this subset. The performance drop in LoRA-tuned LLaMA-2 is attributed to the instability of LoRA tuning, which can occur, especially in unaligned models like LLaMA-2, when compared to full model fine-tuning.
> > > >
> > > > Additionally, when training with text-only data was supplemented by visual instruction data, our experiments demonstrated a 2.5% improvement in model ethics over the Orca baseline. This result is significant, considering that the Orca model is enhanced with advanced reasoning techniques such as chain-of-thought (CoT) prompting and mathematical problem-solving tasks. We hope this addresses your concerns.
> > > >
> > > > **Developing hypothesis**: we develop a hypothesis based on your request:
> > > >
> > > > - **Hypothesis**: *Incorporating visual information into LLMs enhances their ethical reasoning by providing richer contextual grounding and factual knowledge. This visual context supports more informed, contextually appropriate ethical decisions and improves model performance on benchmarks related to ethics and truthfulness.*
> > > >
> > > > - **Supporting Points for the Hypothesis**:
> > > > 1. Enhanced Contextual Grounding: Visual inputs often include contextual clues (e.g., settings, relationships between objects, implied actions) that augment the model's ability to understand and reason about real-world scenarios.
> > > > 2. Integration of Implicit Knowledge: Visual representations can provide implicit information that might not be mentioned explicitly in text but is essential for ethical decision-making. For example, an image depicting a crowded street with individuals following traffic rules can provide visual cues about societal norms and safety considerations that could influence a model’s ethical judgment in relevant questions.
> > > > 3. Bias Mitigation: Visual representations could help mitigate certain biases that text-based models might perpetuate by incorporating diverse, real-world depictions that reflect broader human experiences and cultural practices.
> > > >
> > > > - **Examples**:
> > > > 1. Scenario 1: Imagine an LLM tasked with responding to a question like, “Is it ethical to keep pets in small apartments?” With visual input showing different types of pets in varied living conditions (e.g., a dog in a cramped apartment versus a cat in a spacious room), the model can integrate these visual details to reason more effectively about the ethics of pet care in constrained spaces, assessing well-being beyond textual arguments alone.
> > > > 2. Scenario 2: When asked to evaluate “Is it okay to use automated surveillance in public areas?”, visual content showing public squares, different monitoring technologies, and crowds could provide richer cues. The model could infer how such surveillance may affect individuals' privacy, implicitly learning about societal dynamics that are harder to encode in text alone.
> > > >
> > > > We are happy to incorporate the hypothesis and analysis into our introduction section with your further suggestions.
> > > >
> > > > **RLHF-based baseline**: In the current manuscript, we’ve updated the related work section to include references to the RLHF-based baselines. These newly added baselines are reinforcement learning methods that have been evaluated on Ethics and/or TruthfulQA benchmarks. We stress that these baseline methods utilize specially curated training data and advanced tuning strategies tailored to their specific goals.
> > > >
> > > > However, our primary contribution lies in demonstrating that visual instruction tuning can effectively enhance model performance in terms of truthfulness and ethical perception. To provide further context, we have included the baseline results in our tables, marked in gray to indicate that these figures are not directly comparable to our main results. This distinction ensures clarity while underscoring the unique focus of our approach. We hope these clarifications address your concerns and contribute to a better understanding of the scope and significance of our work.
> > > >
> > > > **Typos**: Apologize for this typo, we have fixed this one.

---

> > > > > ### Comment · Reviewer_ah3g · 2024-11-14
> > > > >
> > > > > Thank you for the additional clarifications.
> > > > >
> > > > > I am still not convinced about the results in Figure 2: you highlight an "average accuracy gain of 1.7%", but I'm not sure how significant this gain really is. For example, looking at LLaMA2-ft, you get a jump from 45.8 to 65.4 after visual instruction tuning (+19.6), which seems very significant, but **tuning on text alone** gives you 65.1, so the additional 0.3 you get from the visual data seems insignificant. In order to make the point that it is the visual instruction tuning data, and not the GPT-4-generated data that changes the evaluation metrics, you could create a table that compares all models, and then compares the relative improvement due to text-only-data vs text-and-image-data. I'm also not sure how much of this difference is due simply to small changes when re-running the evaluation. You could shed some light on this by running transfers five times and report the standard deviation.
> > > > >
> > > > > As for your reported improvements over the **Orca data**, you mention +2.5 when training additionally on the visual instruction-tuning data. What difference do you get when you train on the same data, but text only?
> > > > >
> > > > > Since the central point of your submission is that it is the *visual* instruction tuning data, I think it is really important to make this difference in a controlled setup. As I mentioned in my earlier reply, I would also like to see an improvement of visual instruction tuning where the visual instruction tuning data is **not GPT-4-generated**.
> > > > >
> > > > > Thank you for the elaborated **hypothesis** on how visual input could lead to more ethical answers. I can see your point about additional information conveyed by images that help with ethical decisions, but even in the two examples you mention (about pets in small apartments, and about surveillance in public spaces), I can see how both text or visual data could go both ways (i.e. providing evidence to the "ethical" or "unethical" answer). In my opinion, this argument works a bit better for truthfulness through visual grounding (assuming there are no modified images). One idea to make this clearer is to cherry pick an example from the Ethics dataset that had its answers improved by the visual instruction tuning consistently across most models, and then arguing how visual data could help with that specific example.
> > > > >
> > > > > Adding the **Delphi baseline** provides some idea on what the evaluation numbers on the Ethics benchmark mean. It would be insightful in this context to also add random baselines. Of course it would be better to add baselines that evaluate some of the same models that you have in your table, but I assume that their absence means to there are no other papers examining any of these models under the Ethics benchmark?
> > > > >
> > > > > As for the **baseline on the TruthfulQA** benchmark [(Zhang et al., 2024)](https://arxiv.org/pdf/2410.16843): the authors develop a method that helps retrieval-augmented LLMs to reduce the knowledge conflict between parametric and contextual knowledge. They do not claim that their method improves on TruthfulQA, and in one of the two examples you mention (Vicuna-v1.5-7B), the reported value for TruthfulQA-mc2 is actually a decrease from the baseline. What I would like to see as baselines for the Ethics and TruthfulQA is not an accidental finding of a method designed for another task, but instead a paper where a method is reported that has the goal to improve on truthfulness (from parametric knowledge alone).

---

### Author Response · Authors · 2024-11-03
**General Response**

We thank all reviewers for their thorough and constructive feedback. Specifically, we appreciate that reviewers found our research is valuable for AI alignment and LLM development (NThr, NBFb), the experiments are in-depth and comprehensive (NThr, NBFb, ah3g), our research findings are interesting and potentially beneficial to other areas (NThr, NBFb, ah3g), and our paper is well-written (NBFb). We have carefully considered all the questions raised by the reviewers and address them as follows.

The main changes in the revised submission are as follows:
- We added experiments training LLMs on additional text-only instruction data.
- We included more models, including the latest LLaMA3 series and a newer version of GPT-4-turbo.
- We completed missing results for LoRA tuning and multi-modal benchmarks.
- We provided further details on our motivation, evaluation protocols, and figure statistics.
- We corrected typos and addressed errors in writing and presentation.

In the revised submission, we have marked all changed content in **orange**.

We first present results and analyses of the two major revisions in the general response:

1. **Add Text-only Instruction Tuning Data**: Thank you for your suggestions in adding more tuning data. We incorporated your advice and added Alpaca and Orca text instruction data (downsampled to the same 158K size) for LLM finetuning. We found that the visual instruction tuning data  1) surpasses Alpaca and text-VI data tuned models in most cases, i.e., average 6.7% and 2.8% improvements over Alpaca on two benchmarks, and 0.2%, -1.5% over text-VI; 2) but it lags behind the Orca-tuned model by 1.1% and 16.3% on Ethics and TruthfulQA benchmarks. Considering that the Orca dataset includes Chain-of-Thought (CoT) and complex, nuanced instruction-following data from a diverse array of tasks within the FLAN collection, the observed performance gain is reasonable. The CoT reasoning and complex instruction-following examples in Orca offer richer contextual understanding and problem-solving patterns. And this observation indicates the insight that upgrading text quality within the visual instruction data could further enhance the model’s ethical and truthful reasoning, opening a promising avenue for refining these abilities.
2. **Add Models**: Thank you for your suggestions in adding more models. For closed-API models, we add results of two more recent OpenAI GPT models: GPT-4 Turbo (1205) and GPT-4V Turbo (1205). For open-weight models, we add LLaMA-3 series with 5 more models: LLaMA-3-8B, MM-LLaMA-3-8b-ft,  MM-LLaMA-3-8b-lora, LLaMA-3.1, and LLaMA-3.2. In detail, in the case of OpenAI GPTs, two visually-tuned GPT-4V outperforms their language-only version by an average of 0.85% and 0.55% on Ethics and TruthfulQA-gen tasks. On the LLaMA series, the visually-tuned LLaMA3 models (lora) surpass the original one by 0.25% on two tasks. While LLaMA3.2, the visually-empowered LLM based on LLaMA3.1 consistently surpasses LLaMA3.1 by an average of 1.48% on two tasks.

Then we address specific questions in the individual responses as follows.

---

### Decision · Action_Editor_coFi · 2024-11-25

**Recommendation:** Accept with minor revision

**Comment:**

This submission has been reviewed by three experts, all of whom provided insightful comments. The authors did an excellent job revising the paper, including adding more models and presenting text-only tuning results.

- Reviewers NBFb and NThr consider this submission novel and find its evaluation sufficient. However, Reviewer ah3g raised concerns about whether the improvements in Truthfulness and Ethics arise from the visual data or GPT-generated text data. After carefully reviewing the discussion, I believe Reviewer ah3g raises a valid point. I suggest the authors discuss this concern comprehensively in the revised version. It would also be helpful to incorporate the response from the author-reviewer discussion into the paper. Additionally, I believe Section 3.3 (Tuning on Different Vision-Language Data) supports the claim that visual data contributes to improvements in Truthfulness and Ethics. This submission also has the potential to inspire researchers to investigate the effects of multi-modal training on LLM capabilities beyond existing multi-modal benchmarks.

- Overall, the evaluation is thorough, with numerous results that are well-organized. However, Section 3.4 (Analysis on Multi-Modal Benchmarks) requires some clarification. Specifically, the discussions on “Potential Inconsistencies in Current Multi-Modal Benchmarks” and the “Need for Studying Multi-Modal Alignments” are not very clear. Please elaborate on these points with reference to the results in Tables 5 and 6.

In summary, this submission offers a new perspective on the effects of multi-modal training. The evaluation is comprehensive, and the discussions with reviewers have been constructive. Therefore, I recommend acceptance. Please consider incorporating the above suggestions into the revised version.

**Audience:**

This submission would be of interest to researchers focused on large language models (LLMs) and their alignment with ethical and truthful principles.

**Claims And Evidence:**

The main claim of this submision is *visual instruction tuning, a prevailing strategy to integrate vision knowledge into the LLMs, unexpectedly and interestingly helps models attain both improved truthfulness and ethical alignment in the pure NLP context*.

This submission gives extensive experiments to validate this idea, including varous of LLMs and text-based instruction data. Two reviewers (NBFb and NThr) believe that the new findings are novel and insightful. Yet, Reviewer ah3g still has concern on the GPT-4 generated data VS. visual data: Reviewer ah3g stresses the importance of evaluating visual instruction tuning data that is not GPT-4-generated to strengthen the submission’s claims.

---

> ### Author Response · Authors · 2024-12-20
>
> Thank you for your feedback and recognition of our work! We have revised the manuscript based on your valuable suggestions:
>
> 1. Expanded discussion on data: We have integrated the author-reviewer discussions into the camera-ready version, including the newly developed hypothesis and an analysis of GPT-generated data.
> 2. Improved Section 3.4: Following your suggestions, we have enhanced Section 3.4. Specifically, we provide a more detailed comparison of aligned and unaligned LLMs on multi-modal benchmarks and emphasize our claim that “MLLMs still face unique challenges in the multi-modal context, particularly when dealing with corrupted or imperfect visual inputs,” with appropriate references.
>
> We believe these changes strengthen the manuscript, and we appreciate your constructive input.